# Remote Heart Rate Estimation Based on Transformer with Multi-Skip Connection Decoder: Method and Evaluation in the Wild

**DOI:** 10.3390/s24030775

**Published:** 2024-01-25

**Authors:** Walaa Othman, Alexey Kashevnik, Ammar Ali, Nikolay Shilov, Dmitry Ryumin

**Affiliations:** 1St. Petersburg Federal Research Center of the Russian Academy of Sciences (SPC RAS), 199178 St. Petersburg, Russia; walaa_othman@itmo.ru (W.O.); nick@iias.spb.su (N.S.); ryumin.d@iias.spb.su (D.R.); 2Information Technology and Programming Faculty, ITMO University, 191002 St. Petersburg, Russia; ammarali32@itmo.ru

**Keywords:** remote heart rate estimation, vital signs, machine learning, video analysis

## Abstract

Heart rate is an essential vital sign to evaluate human health. Remote heart monitoring using cheaply available devices has become a necessity in the twenty-first century to prevent any unfortunate situation caused by the hectic pace of life. In this paper, we propose a new method based on the transformer architecture with a multi-skip connection biLSTM decoder to estimate heart rate remotely from videos. Our method is based on the skin color variation caused by the change in blood volume in its surface. The presented heart rate estimation framework consists of three main steps: (1) the segmentation of the facial region of interest (ROI) based on the landmarks obtained by 3DDFA; (2) the extraction of the spatial and global features; and (3) the estimation of the heart rate value from the obtained features based on the proposed method. This paper investigates which feature extractor performs better by captioning the change in skin color related to the heart rate as well as the optimal number of frames needed to achieve better accuracy. Experiments were conducted using two publicly available datasets (LGI-PPGI and Vision for Vitals) and our own in-the-wild dataset (12 videos collected by four drivers). The experiments showed that our approach achieved better results than the previously published methods, making it the new state of the art on these datasets.

## 1. Introduction

In recent years, there has been a noticeable surge in stress levels in our daily lives, underscoring the crucial need to proactively monitor overall human health. The assessment of vital signs, such as respiratory rate, heart rate, blood pressure, and oxygen saturation levels, has garnered increasing attention among researchers. These vital signs serve as pivotal indicators in evaluating the holistic well-being of the human body and play crucial roles in various applications, including the detection of fatigue and stress levels [1,2], disease diagnosis [3], and assessments related to meditation and fitness.

Various systems have been devised to measure these vital signs, encompassing devices like blood pressure monitors, oximeters, and electrocardiogram machines. While these systems deliver precise measurements, their requirement for direct contact with the subject makes them more suited for clinical settings. However, this feature becomes a drawback, causing discomfort, when the subject is engaged in physical exercises or driving.

An alternative approach involves leveraging cameras to remotely estimate vital signs [4,5,6]. This is achieved by capturing variations in light reflected from the skin, induced by physiological processes such as the heartbeat and breathing. This non-contact methodology not only offers convenience but also opens up possibilities for unobtrusive health monitoring, proving especially valuable in situations where direct contact is impractical or uncomfortable.

However, research on the topic of remote heart rate detection is usually conducted in good lighting and shaking conditions that do not reflect settings like vehicle cabins and streets. In this paper, we propose a new method for remote heart rate detection and evaluate it using our own dataset of drivers in vehicle cabins recorded in the wild. The contributions of this paper can be summarized as follows:We investigate the best feature extractor to capture heart rate spatial information from the face.We investigate the optimal number of frames to achieve better accuracy in captioning the global features.We propose a new method based on the vision transformer architecture to estimate the heart rate with a novel pre-processing technique.We propose a novel approach of treating a single regression value problem by extracting intervals, which optimizes the performance and opens the way for using new techniques like the intersection over union of the intervals and complex loss functions to optimize interval values.We evaluated the proposed approach on three different datasets (including our own dataset recorded in the wild). In comparison with the previously published methods, our approach achieved the highest accuracy.

The rest of the paper is organized as follows. Section 2 summarizes the state-of-the-art methods used to estimate the heart rate. Section 3 introduces the proposed method in detail. The experiments conducted are presented in Section 4. Finally, the conclusion is provided in Section 5.

## 2. Related Work

Remote heart rate estimation has recently drawn the attention of researchers due to its importance in health monitoring without causing discomfort to the subject. Several methods have been used to extract heart rate information from facial videos using mathematical approaches and signal processing techniques [7,8,9,10]. In [7], the authors extracted heart rate information by highlighting the green channel only, since it contained the highest photoplethysmography (PPG) signal-to-noise ratio. Refs. [8,11] introduced different mathematical reflection models of the skin that were used to build models for extracting remote PPG signals. Ref. [10] applied the method proposed in [8] to five sub-regions (the face, forehead, right and left cheek, and nose) to extract remote PPG signals, followed by fast Fourier transform and bandpass filtering to obtain a power spectrum density for each sub-region. A sorting function was applied to each region to determine which ROI would be used to extract the heart rate. With the advances in computer vision and machine learning techniques, more accurate methods have been proposed. Some researchers have used deep convolutional neural networks to estimate the heart rate from videos [12,13,14,15,16,17]. Ref. [12] improved the robustness of the heart rate estimation model under different motion and illuminance conditions by using a skin reflection model and an appearance information attention mechanism. Refs. [13,14] proposed a two-branch model: appearance and movement. The input to the former branch was the current frame, while the input to the latter was the normalized difference between the current and the next frames. The output of these branches was then combined to estimate the heart rate. Ref. [17] introduced a hybrid-CAN-RNN framework by adding a bidirectional GRU on top of the hyprid-CAN model proposed in [13]. Ref. [16] proposed aggregating the remote PPG signals from multiple skin areas to improve reliability. Other researchers proposed using the long short-term memory (LSTM) layer to capture the features between frames [18,19]. Ref. [18] proposed two branches of neural networks: the first one used 3D pooled convolutional layers with several consecutive frames as an input, while the second branch used a 2D convolutional layer with LSTM layers, and the input to this branch was one frame at a time. The outputs of the branches were then combined to estimate the heart rate. Ref. [19] proposed a model consisting of two LSTM networks: one to estimate the heart rate sampling point, and the other to predict the signal quality. These two values were then fed into an attention-based model to estimate the average heart rate. A video transformer was proposed in [20]. The authors’ main contribution was calculating the loss in the frequency domain instead of the time domain. Ref. [21] introduced a self-supervised framework. A face extractor was first used to detect the face, and then a remote PPG estimator based on a 3D-CNN was used to extract the PPG signal. The authors proposed a new way to augment the dataset by stretching and squeezing the video. When the video is squeezed, the heart rate should increase, and vice versa. Ref. [22] proposed an end-to-end architecture using 3D depth-wise separable convolution layers with residual connections for heart rate estimation from videos.

The existing heart rate estimation models based on deep learning methods use LSTM-based models, convolutional models, attention mechanisms, or a combination of the above to predict the heart rate value. In our paper, we present an architecture based on a vision transformer with a multi-skip connection decoder. Unlike previously published methods, our model takes the spatial features from five different layers of the feature extractor, allowing the model to capture diverse features. These features are then fed into BiLSTM layers, followed by 1D Conv and linear layers outputting five different arrays. Each array contains the predicted minimum, maximum, and average HR in the selected window. These values are then combined together using the weighted average method to predict the final heart rate value.

## 3. Proposed Approach

### 3.1. General Approach

In this section, we describe the developed approach for predicting heart rate. Figure 1 shows the approach based on the transformer architecture with a multi-skip connection decoder.

To predict the heart rate, we used a sliding window with a length of 15 frames and a stride of 15 frames. The length of the sliding window was chosen based on our experiments (more details are described in Section 4.3), while the stride was chosen to reduce the correlation between the training data to the minimum. We propose processing the input frames using 3DDFA_V2 [23,24] to extract 68 facial landmarks: 17 for the face, 10 for the eyebrow, 9 for the nose, 10 for the eyes, and 22 for the mouth. 3DDFA_V2 is the SOTA on the Florence dataset for 3D face reconstruction (in our case, the person could move their head, and 3DDFA_V2 could keep providing the facial landmarks even when the face was completely turned to the left or right due to the fact that it provided 3D coordinates). Figure 2 shows a 3D face reconstruction of a person from our dataset turning his head to the side produced using 3DDFA_V2. In addition, according to [25], 3DDFA_V2 achieved more robust and accurate performance in different movement conditions compared with OpenFace 2.0 [26] and MediaPipe [27].

To detect the forehead area, we added two more landmarks based on the left-most and right-most eyebrow points. The obtained landmarks were used to segment the face by keeping only the pixels inside the landmarks and filling all other pixels with zeros. The obtained images were then cropped and resized into 224 by 224 pixels. We applied normalization to the images before feeding them into the model.

In this paper, we propose two architectures for heart rate estimation. The first one consists of a feature extractor based on the vision transformer (VIT) [28] with BiLSTM and a linear layer with one output that represents the predicted HR. The second one takes the outputs’ connections from five different blocks of the feature extractor and feeds them into several BiLSTM layers followed by 1D convolution and linear layers with three outputs each. The three outputs represent the minimum, maximum, and average HR in the sliding frame. We trained the first model with the mean absolute error (MAE) as the loss function. The MAE was calculated between the predicted and the real heart rate values. During the training phase of the second model, the five outputs were averaged together, and the loss function was calculated as the MAE of the three values. In the testing phase, the weighted average was used to calculate the predicted heart rate from the three obtained values. More details of the proposed architectures are provided in Section 3.2.

### 3.2. Proposed Architectures

In this section, we introduce the architectures used to predict the heart rate. Our first model architecture is shown in Figure 3.

The model consisted of a VIT feature extractor to extract the spatial features followed by a BiLSTM layer to extract the global features and three linear layers used as an estimator to estimate the heart rate value. The BiLSTM layer was used to handle the features of multiple frames (we chose 15) since LSTM can selectively remember or forget information according to its importance, though an attention mechanism could be used here instead. LSTM maintains the order of embeddings, with the output at time *t* depending on the output at t−1; thus, because of its sequential property, it is more effective for video analysis. The feature extractor was chosen based on experiments (more details are provided in Section 4.2). The MAE (see Equation (Equation 1)) was used as the loss function.
(1)$$MAE=\frac{\sum_{i}|HR_{i}-\hat{HR_{i}}|}{N},$$
where $HR_{i}$ is the heart rate ground truth of sample *i* in the dataset, $\hat{HR_{i}}$ is the predicted heart rate of sample *i*, and *N* is the total number of samples in the dataset. To evaluate the model’s performance, in addition to the MAE metric, the root mean square error (RMSE) given in Equation (Equation 2) was used.
(2)$$RMSE=\sqrt{\frac{1}{N}\sum_{i=1}^{N}{\left(HR_{i}-\hat{HR_{i}}\right)}^{2}}$$

The model showed good results, with an MAE equal to 2.67 beats per minute (BPM) on the LGI-PPGI dataset and 9.996 BPM on the Vision for Vitals (V4V) dataset.

In order for the model to be able to capture different features, we modified the architecture as shown in Figure 4.

The modifications included taking multi-skip connections from five different blocks of the VIT feature extractor. As we know, each block attends to different features. This allows the model to obtain diverse spatial feature maps from the image. In other words, the spatial features taken from the five different blocks differ from each other. Since the Vit_small_patch16_224 feature extractor used in our model consists of 12 blocks, we chose to take the connections from five blocks that were of different depths. From Figure 4, we can see that the connections were taken from blocks 0, 3, 6, 9, and 11. We included a distance of three blocks to avoid redundancy, as feature maps from blocks that are relatively close in depth are similar to each other. Each connection was fed into a block consisting of a BiLSTM layer to capture the global features, followed by batch normalization, a 1D convolution layer, and a linear layer to estimate three values: the minimum, maximum, and average heart rate. During the training phase, the outputs from the five blocks were averaged, and the MAE loss function was used as the criterion. In the prediction phase, the three values were combined as shown in Equation (Equation 3) to estimate the final heart rate value.
(3)$$\hat{HR_{i}}=0.25*\frac{\hat{MinHR_{i}}+\hat{MaxHR_{i}}}{2}+0.75*\hat{AvgHR_{i}},$$
where $\hat{MinHR_{i}},\hat{MaxHR_{i}},$ and $\hat{AvgHR_{i}}$ are the outputs of the model (the predicted minimum, maximum, and average values of sample *i*, respectively), and $\hat{HR_{i}}$ is the final estimation of the heart rate of sample *i*. The weights were chosen empirically. We used multi-skip connection since the vision transformer split the face image into patches and then transformed it into a sequence. The self-attention mechanism considered the main feature, which in our case was the skin color affected by illumination, camera type, and other parameters. Moreover, the multiple outputs of the model (max and min heart rate values) depended on the embedding of two different patches; therefore, using multi-skip connection helped the model to converge better and to consider the high score attention at multiple levels.

## 4. Experiments

The Pytorch 1.10.2+cu113 framework was used to implement our models. We used an Nvidia RTX 3090 GPU (Nvidia, China) to train and validate the models. We trained and validated both models on two public datasets: LGI-PPGI and V4V. We also tested the trained models on our own dataset to study their generalization in different environments and with different cameras. The hyper-parameters were tuned using Ray Tune [29]. We ran 24 experiments on 40% of the LGI-PPGI dataset to find the optimal learning rate/batch size combination within the ranges [$1\times 10^{-6}$, $1\times 10^{-1}$] and [2,4,8] for the learning rate and batch size, respectively. Figure 5 shows the results of the experiments.

### 4.1. Datasets

We used three facial datasets with heart rate annotation: two publicly available ones recorded in an indoor environment and one collected from four drivers in real driving scenarios. The first dataset was Vision for Vitals (V4V) [30,31]. It consists of frame-aligned annotated facial videos of 179 subjects from diverse racial groups aged between 18 and 66. The videos were recorded with a frame rate of 25 frames per second (FPS) and a resolution of 1040 × 1392 pixels. The dataset was split into training, validation, and testing sets consisting of 724, 276, and 358 videos, respectively, representing 53.31%, 20.23%, and 26.36% of the dataset.

In this paper, we conducted the experiments only on the validation dataset, as the ground truth for the testing dataset was not provided by the dataset owners.

The second dataset was LGI-PPGI [32], collected using a Logitech HD C270 webcam with subjects aged between 25 and 42 years in four different scenarios: talking, rotation, resting, and gym. The videos were recorded with a frame rate of 25 FPS, while the PPG signal was recorded with a sampling rate of 60 HZ.

The final dataset was our dataset [33], which consisted of 12 videos with an average video length of 20 min (907,437 frames in total). The videos were taken using smartphone cameras in different illumination and movement conditions with a frame rate of 60 FPS and depicted 4 drivers in real driving scenarios. We measured the heart rate using a Xiaomi MI Smart Band 3, China.

We used this dataset for testing the generalization capability of the developed models. Figure 6 shows the distribution of heart rate in our dataset.

### 4.2. Choosing the Feature Extractor

Choosing the best feature extractor that can capture spatial information is mandatory for the heart rate estimation problem. We ran several experiments on different feature extractors in order to choose the best one to use in our models. Experiments were conducted using the architecture shown in Figure 3 with the replacement of vit_small_patch16_224 in each experiment. The dataset used was LGI-PPGI, with 25 frames per sample. Table 1 shows a comparison between different feature extractors.

Large models (e.g., vit_base_patch16_224) were tested, but they showed worse results as they tended to overfit quickly on the dataset. For this reason, we excluded them from the table. One can see (Table 1) that the vit_small_patch16_224 model showed the best result, so we used it in our models.

### 4.3. Determining the Optimal Number of Frames

Determining the number of frames per sample needed to capture the global features is very important for estimating the heart rate. We performed several experiments to choose the optimal number of frames. The experiments were conducted using the model shown in Figure 3 with vit_small_patch16_224 as the feature extractor. For each experiment, the model was trained on 20% of the V4V training dataset and tested on 20% of the V4V testing dataset. Table 2 shows the MAE on the validation dataset. The optimal number of frames per sample was 15 frames.

### 4.4. Results on Our Dataset

As we mentioned earlier, we further tested our trained models on a completely new dataset, which was recorded using different cameras in real driving scenarios to evaluate the model generalization. Since our dataset was collected with a frame rate of 60 FPS and the models were trained with a frame rate of 25 FPS, we first down-sampled all the videos to 25 FPS before feeding them into the models. Table 3 shows the results of our models, which were trained on the V4V dataset and tested on our dataset.

We show in Table 3 that both models had an MAE less than 14.4 BPM, which is a good result, taking into consideration that our dataset was recorded in varying circumstances of illumination and inside a moving vehicle.

### 4.5. Comparison with Previous Works for HR Estimation

Since our models were applied to estimate the heart rate from RGB videos and we used the LGI-PPGI and V4V datasets, we compared our models with previous models trained on these two datasets.

Table 4 shows a comparison between our models and the previously published work on the LGI-PPGI dataset. As shown in the table our models achieved better results on this dataset.

Similarly, Table 5 shows the comparison on the V4V dataset. The comparison showed that our results were better. Here, we would like to mention that the authors of [17,21,22] also applied the V4V dataset, but we could not compare our results with theirs as they only published their results for the testing dataset and not for the validation dataset, and they did not provide an open-source code for their methods.

The model with the multi-skip connections showed better results when the heart rate range in the sample was wide. For example, in the case of the V4V dataset, the range between the minimum and maximum heart rate values in one sample (15 frames) could exceed 20 BPM. Furthermore, we calculated the number of parameters and the number of FLOPS during inference for our models. For the first architecture, #FLOPS = 8.156T and #parameters = 21.799M; for the second architecture, #FLOPS = 14.8T and #parameters = 22.419M. Unfortunately, none of the previously published works provided these numbers in their papers for comparison.

## 5. Conclusions

In this paper, we proposed two models to estimate the heart rate from RGB videos. Both models achieved a more accurate estimation of the heart rate than the previously published methods on the two tested datasets: LGI-PPGI and V4V. Furthermore, we tested the trained models on our own dataset recorded in the wild using different cameras and different lightning and driver movement conditions in a moving vehicle. This dataset allowed us to evaluate the proposed models’ generalization. The proposed models also showed good results on this dataset, with an MAE less than 14.4 BPM. In addition, we investigated the best number of frames per sample to capture the global features and the best feature extractor to extract the spatial features.

## Figures and Tables

**Figure 1 sensors-24-00775-f001:**
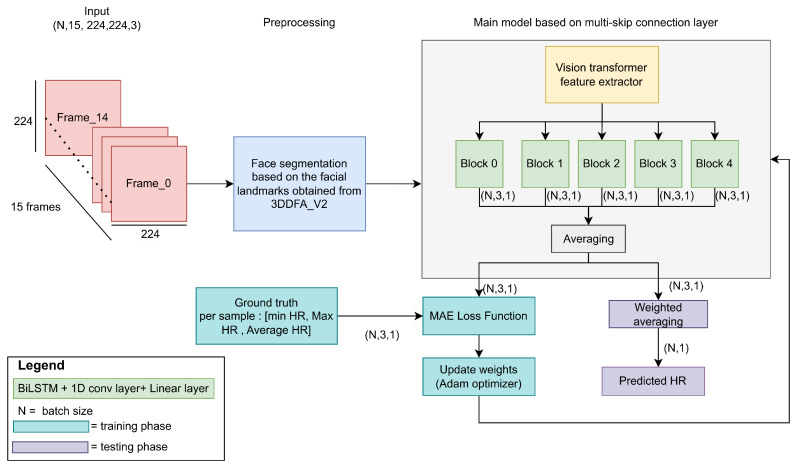
The developed approach based on multi-skip connection decoder.

**Figure 2 sensors-24-00775-f002:**
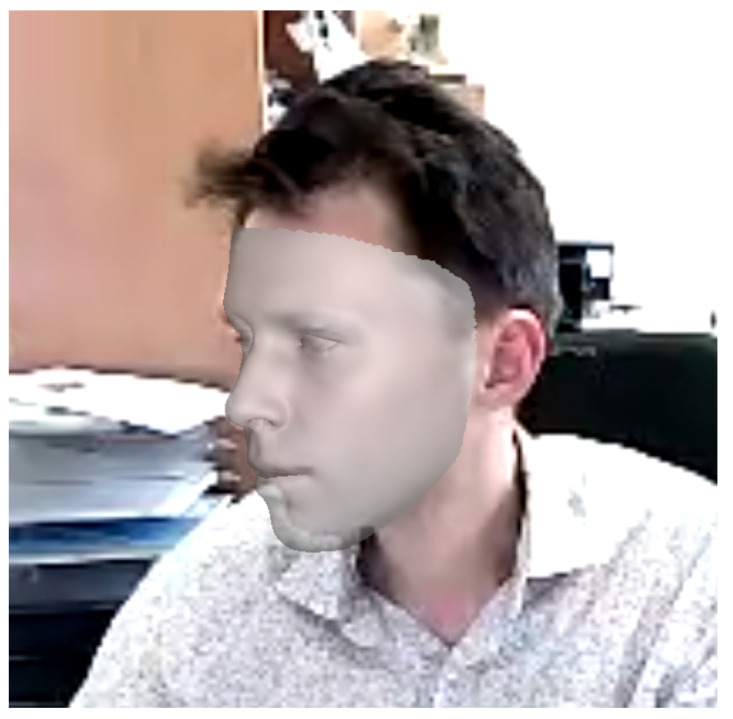
3DDFA_V2 face reconstruction result of a person turning his head to the side.

**Figure 3 sensors-24-00775-f003:**
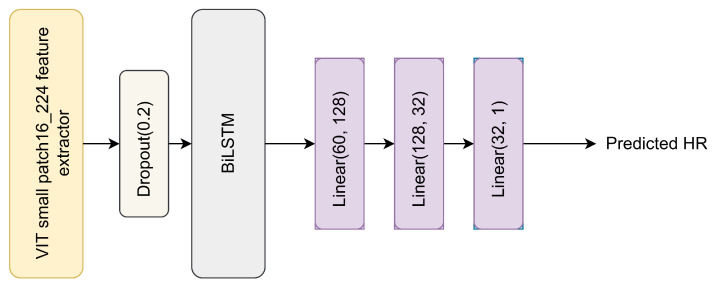
The first proposed model architecture.

**Figure 4 sensors-24-00775-f004:**
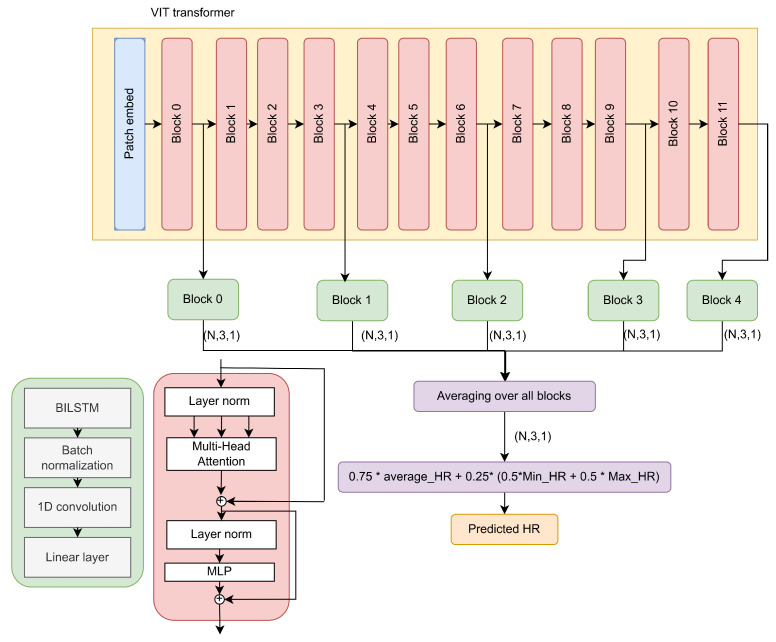
The proposed architecture based on VIT transformer with multi-skip connection decoder. The connections were taken from the output of blocks 0, 3, 6, 9, and 11 and were then fed into a block consisting of a BiLSTM layer followed by batch normalization, 1D convolution, and a linear layer. The output of each block was 3 numbers representing the predicted mean, max, and min heart rate. These values were then averaged to produce the final predicted value.

**Figure 5 sensors-24-00775-f005:**
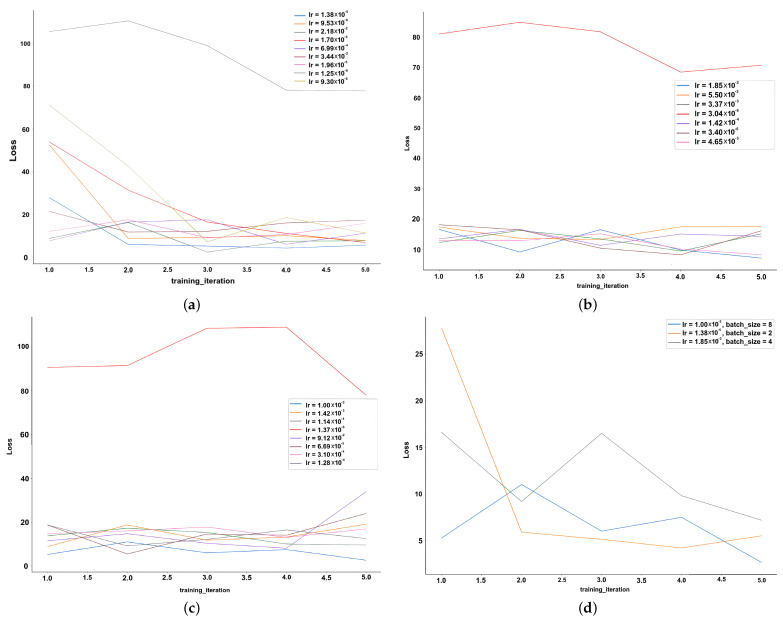
The results of the experiments for tuning the hyper-parameters, (**a**) Batch size = 2, (**b**) Batch size = 4, (**c**) Batch size = 8, and (**d**) shows the best combinations.

**Figure 6 sensors-24-00775-f006:**
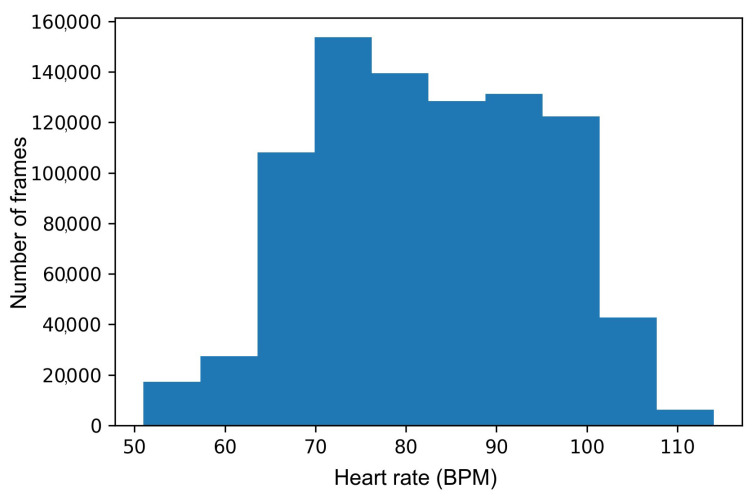
The heart rate distribution in our dataset.

**Table 1 sensors-24-00775-t001:** The results of different feature extractors on LGI-PPGI dataset. The best score is highlighted in boldface.

Method	MAE
Efficientnet_b1_ns [34]	7.26
Efficientnet_b3_ns [34] (frozen 80%)	5.57
Swin_base [35]	20.38
Swin_transformer_large_224 [35]	5.83
Swin_transformer_large_224 (frozen 50%)	3.75
Resnet18 [36]	12.3
Resnet50 [36]	20.35
**vit_small_patch16_224 [28]**	**3.34**

**Table 2 sensors-24-00775-t002:** The MAE of 20% of the V4V validation dataset using the model trained on 20% of the training dataset with different numbers of frames per sample. The lowest MAE is highlighted in boldface.

Number of frames	MAE
5 frames	11.21
**15 frames**	**10.34**
25 frames	11.13
40 frames	10.95
75 frames	10.81
90 frames	10.74

**Table 3 sensors-24-00775-t003:** MAE of heart rate estimation by our models trained on V4V dataset and tested on our dataset (unit: BPM).

Model	MAE
First proposed model	14.33
Second proposed model with multi-skip connection	14.09

**Table 4 sensors-24-00775-t004:** The comparison results for the LGI-PPGI dataset. The best score is highlighted in boldface.

Method	MAE	RMSE
Green [7]	5.53	-
ICA [9]	5.81	-
CHROM [11]	5.05	-
POS [8]	7.87	21
1D-CNN [15]	8.44	-
LSTM-rPPG [19]	7.12	-
SQA-rPPG [19]	7.05	-
**Our model**	**2.68**	**5.69**
**Our model with designed encoder**	**4.06**	**6.75**

**Table 5 sensors-24-00775-t005:** The comparison results for the V4V dataset. The best score is highlighted in boldface.

Method	MAE (Validation Dataset)	RMSE (Validation Dataset)
Green [7]	16.5	21.4
POS [8]	17.3	21.2
ICA [9]	13.9	21.2
DeepPhys [12]	13.6	18.1
TS-CAN [13]	11.7	17.8
Kossack et al. [10]	10.5	13.648
Video Transformer model [20]	10.3	16.1
**Our model**	**9.996**	**13.4**
**Our model with designed decoder**	**9.8**	**12.8**

## Data Availability

Data are contained within the article.

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
