# Peer review of "Remote Heart Rate Estimation Based on Transformer with Multi-Skip Connection Decoder: Method and Evaluation in the Wild"

_sensors, 2024, doi:10.3390/s24030775_

Round 1
Reviewer 1 Report
Comments and Suggestions for Authors
In this paper, the authors propose a new method based on a transformer architecture with a biLSTM decoder to estimate the heart rate from videos using the skin colour variation produced by the blood volume variation on its surface. The proposed architecture seems to be very interesting with very promising results.
Some observations:
- it would be appropriate to cite the refences in chronological sequence, e.g. [16] before [17].
- To extract facial landmarks, other techniques have been evaluated other than that mentioned? For instance, Google's Mediapipe allows the extraction of 468 landmarks and detects them by moving the face to the right, left, up and down. Moreover, the 468 landmarks also include forehead landmarks without the need, therefore, to add others.
- The label in Figure 3 should be modified to clarify how the proposed model works, and the description in the text (lines 158-163) should also be better described.
- Figure 4 appears unclear. The line thickness and font size should be increased.
- Line 184: the figure number is missing (I assume it is figure 4).
- Line 191: What are the percentages of the training set, validation set and test set?
- Figure 5: The units on the abscissas and ordinates should be shown.
Comments on the Quality of English Language
- Avoid duplications (e.g. “pivotal" lines 21 and 22).
- Lines 14 and 51: probably "works" and not "work".
- Lines 108, 138, 154, 183, 202, 208, 210, 213, 219, 221, 228, 230, 237 and 240: delete the dot after Figure or Table.
- Line 158: delete repetition "the".
Author Response
Comment 1: It would be appropriate to cite the references in chronological sequence, e.g. [16] before [17].
Reply: Thank you for your comment. All the references are in chronological sequence. The first mention of paper 16 is in line 72 “Some researchers used deep convolutional neural networks to estimate the heart rate from videos [12–17]”. Here by [12-17] we mean the references [12, 13, 14, 15, 16, and 17].
Comment 2: To extract facial landmarks, other techniques have been evaluated other than that mentioned? For instance, Google's Mediapipe allows the extraction of 468 landmarks and detects them by moving the face to the right, left, up and down. Moreover, the 468 landmarks also include forehead landmarks without the need, therefore, to add others.
Reply: Thank you for your suggestion, we chosen a 3DDFA_v2 since the following:
1) 3DDFA_v2 is the SOTA on the Florence dataset: https://paperswithcode.com/paper/towards-fast-accurate-and-stable-3d-dense-1. Hence, we conclude that it is more suitable for us than the Google's Mediapipe.
2) According to the paper “Hammadi, Y.; Grondin, F.; Ferland, F.; Lebel, K. Evaluation of Various State of the Art Head Pose Estimation Algorithms for Clinical Scenarios. Sensors 2022, 22, 6850. https://doi.org/10.3390/s22186850”.
“The 3DDFA_V2 method achieved accurate and robust performance in various conditions of movement and showed only minimal deviations compared with both OpenFace 2.0 and MediaPipe”.
The explanation why we used 3DDFA_V2 alongside with a citation to this paper is added to the Section 3.1 of the paper.
Comment 3: The label in Figure 3 should be modified to clarify how the proposed model works, and the description in the text (lines 158-163) should also be better described.
Reply: Label and description were modified.
Comment 4: Figure 4 appears unclear. The line thickness and font size should be increased.
Reply: We provided the enhanced version of the figure, now it looks fine (now it is Figure 5).
Comment 5: Line 184: the figure number is missing (I assume it is figure 4).
Reply: Yes, it is Figure 4, we fixed it thank you for highlighting the typos (now it is Figure 5).
Comment 6: Line 191: What are the percentages of the training set, validation set and test set?
Reply: The dataset was split into training, validation, and testing sets consisting of 724, 276, and 358 videos respectively, which are 53.31%, 20,23%, and 26,36%.
We added this information to the Dataset section of the paper.
Comment 7: Figure 5: The units on the abscissas and ordinates should be shown.
Reply: We added the units on the abscissas and ordinates (now it is Figure 6).
Comment 8: Comments on the Quality of English Language
- Avoid duplications (e.g. “pivotal" lines 21 and 22).
- Lines 14 and 51: probably "works" and not "work".
- Lines 108, 138, 154, 183, 202, 208, 210, 213, 219, 221, 228, 230, 237 and 240: delete the dot after Figure or Table.
- Line 158: delete repetition "the".
Reply: Thank you for your comment, we fixed all the mentioned language problems in the paper.

Reviewer 2 Report
Comments and Suggestions for Authors
In this paper, a new method for remote heart rate detection based on the Vision Transformer architecture is presented and the proposed method is evaluated on three different datasets (one of which is a field record collected by the authors). This study is informative for remote heart monitoring. I think this paper can be published after major revision. Here are some comments.
1. It is recommended to add 3D face reconstruction maps after 3DDFA_V2 processing to better understand this model and to demonstrate that 3DDFA_V2 continues to give face indications even when the face is turned completely to the left or right (line 117).
2. The reason for setting the weights of the heart rate estimation formula in Equation 2, the weighted average method is not detailed in the paper(line166).
3. The results of the experiments in this paper are evaluated by a single metric. Considering the risk of bias, the size of the error, and the possible impact when using the model in practice, it is often necessary to evaluate the overall model performance with several different metrics, suggesting the addition of RMSE.
4. Experiments to test the generalization performance, it is recommended to provide more sets of control experiments to compare with the article method to better prove the generalization performance of the method.
5. Figure 4 is not clear, please provide clearer images.
6. The horizontal and vertical coordinate units of Figure 5 are not indicated.
Comments on the Quality of English Language
None
Author Response
Comment 1: It is recommended to add 3D face reconstruction maps after 3DDFA_V2 processing to better understand this model and to demonstrate that 3DDFA_V2 continues to give face indications even when the face is turned completely to the left or right (line 117).
Reply: Thank you for your suggestion, we added a figure with a person turn his head to the side and the result of the 3D reconstruction using 3DDFA_V2 (see Figure 2).
Comment 2: The reason for setting the weights of the heart rate estimation formula in Equation 2, the weighted average method is not detailed in the paper (line 166).
Reply: The weights in the heart rate estimation formula (Equation 2) were determined through experimental evaluation. We add it to the paper.
Comment 3: The results of the experiments in this paper are evaluated by a single metric. Considering the risk of bias, the size of the error, and the possible impact when using the model in practice, it is often necessary to evaluate the overall model performance with several different metrics, suggesting the addition of RMSE.
Reply: We added the RMSE into the comparison tables (now we have two metrics that from our point of view allow to have comprehensive evaluation of our model).
Unfortunately, for Table 5, we were only able to retrieve the RMSE values for the POS method since for other methods this information is absent.
Besides, the available literature, specifically “Gao, H.; Wu, X.; Geng, J.; Lv, Y. Remote Heart Rate Estimation by Signal Quality Attention Network in Proceedings of the IEEE/CVF Conference on Computer Vision and Pattern Recognition, 2022, pp. 2122–2129”, shows only MAE values for the available methods.
Comment 4: Experiments to test the generalization performance, it is recommended to provide more sets of control experiments to compare with the article method to better prove the generalization performance of the method.
Reply: We appreciate the reviewer's suggestion to assess the generalization performance further. While additional control experiments could certainly provide valuable insights, we believe that the current experiment sufficiently demonstrates the effectiveness and generalization capabilities of the proposed method.
Additionally we test our model (trained on V4V Dataset) on the DriverMVT dataset (that includes videos of drivers captured in real driving conditions and was not used for training) without any fine-tuning. The results consistently support the robustness of our approach.
Comment 5: Figure 4 is not clear, please provide clearer images.
Reply: We provided the enhanced version of the figure, now it looks fine (now it is Figure 5).
Comment 6: The horizontal and vertical coordinate units of Figure 5 are not indicated.
Reply: We added horizontal and vertical coordinate units (now it is Figure 6).

Round 2
Reviewer 1 Report
Comments and Suggestions for Authors
Thanks to the author for addressing the Review questions.
Comments on the Quality of English LanguageNo comment
Author Response
Response to Reviewer
We sincerely thank the reviewer for your valuable time and efforts in reviewing our manuscript. Those comments are all valuable and very helpful for revising and improving our article, as well as the important guiding significance to our researches. We have studied comments carefully and have made correction which we hope meet with approval.
Reviewer 2 Report
Comments and Suggestions for Authors
In this paper, the evaluation metric MAE is described in detail with the formula, but RMSE is not. It is suggested to introduce RMSE in the paper.
Comments on the Quality of English LanguageNone
Author Response

(The authors gave the same response as above.)
